# Prediction of Technological Change under Shared Socioeconomic Pathways and Regional Differences: A Case Study of Irrigation Water Use Efficiency Changes in Chinese Provinces

**Aijun Guo [1], Daiwei Jiang [1], Fanglei Zhong [1],\*, Xiaojiang Ding [1], Xiaoyu Song [2],\*, Qingping Cheng [2] , Yongnian Zhang [1] and Chunlin Huang [2,3]**

[1]   School of Economics, Lanzhou University, Lanzhou 730000, China; guoaj@lzu.edu.cn (A.G.); jiangdw18@lzu.edu.cn (D.J.); dingxj17@lzu.edu.cn (X.D.); zhangyongnian12@163.com (Y.Z.)
[2]   Northwest Institute of Eco-Environment and Resources, Chinese Academy of Sciences, Lanzhou 730000, China; qpchengtyli@foxmail.com (Q.C.); huangcl@lzb.ac.cn (C.H.)
[3]   Key Laboratory of Remote Sensing of Gansu Province, Heihe Remote Sensing Experimental Research Station, Northwest Institute of Eco-Environment and Resources, Chinese Academy of Sciences, Lanzhou 730000, China
\*   Correspondence: zfl@lzu.edu.cn (F.Z.); songxy@llas.ac.cn (X.S.)

**Abstract:** Technological changes in water use efficiency directly influence regional sustainable development. However, few studies have attempted to predict changes in water use efficiency because of the complex influencing factors and regional diversity. The Chinese Government has established a target of 0.6 for the effective utilization coefficient of irrigation water, but it is not clear how the coefficient will change in different provinces in the future. The purpose of this study is to predict irrigation water use efficiency changes using a conditional convergence model and combined with the shared socioeconomic pathways (SSPs) scenario settings and hydro-economic (HE) classification to group 31 Chinese provinces by their different economic and water resources conditions. The results show that the coefficient exponentially converges to 0.6 in half the provinces under SSP1 (sustainability), SSP2 (middle of the road), and SSP5 (conventional development) by 2030, whereas SSP3 (fragmentation) and SSP4 (inequality) are generally inefficient development pathways. HE-3 provinces (strong economic capacity, substantial hydrological challenges) achieve the greatest efficiency improvements (with all coefficients above 0.6), and SSP1 is a suitable pathway for these provinces. HE-2 provinces (strong economic capacities, low hydrological challenges) have relatively low efficiency because they lack incentives to save water, and SSP1 is also suitable for these provinces. For most HE-1 provinces (low economic capacity, low hydrological challenges), the coefficients are less than 0.6, and efforts are required to enhance their economic capacity under SSP1 or SSP5. HE-4 provinces (low economic capacity, substantial hydrological challenges) would improve efficiency in a cost-efficient manner under SSP2.

**Keywords:** technological change prediction; shared socioeconomic pathways (SSPs); hydro-economic classification (HE); conditional convergence model; irrigation water use efficiency; Chinese provinces

## 1. Introduction

When considering the issue of the sustainability of economic growth, the constraints of natural resources and climate change are always present, and the interaction between climate change and economic growth is of increasing concern [1,2]. Technological change, including changes in the

utilization efficiency of natural resources, is a possible way to solve the current dilemma of sustainable development [3,4], because it has a direct impact on the total scale of natural resource utilization, and thus directly influences whether the regional social economy is in a sustainable development state [5,6]. Technological change in agriculture irrigation water use efficiency is of fundamental significance for solving water scarcity and increasing crop productivity, and achieving highly efficient irrigation is crucial to balancing water resources input and sustainable agricultural economic growth [7–9].

The theory of technological change generally distinguishes two types of technological change: technological catch-up and technological diffusion. Technological catch-up concerns the knowledge production function and occurs through the mechanism of learning by doing. It requires continuous additional capital inputs and manufacturers, and the labor force must constantly learn and master new skills in the production process, which brings about extensive progress in social productivity [3,10–12]. Technological diffusion is brought about by technological transmission and is mainly realized through open trade, technology transfer, information flows, and spatial spillover effects [13–16]. It is generally believed that technological change will gradually converge to an optimal efficiency level at a certain stage of development. At the same time, the speed of improvement of an advanced region is slower than that of a backward region, which is relevant to the distribution of a conditional convergence model. In this study, the conditional convergence model refers to an exponential model that reflects the long-term changes in technological efficiency of economies with similar structural characteristics [17–20].

The coefficient of the effective utilization of irrigation water is a comprehensive technological efficiency indicator that reflects the quality of irrigation projects, the level of irrigation technology, and the level of water management, which generally refers to the ratio of the amount of water that can be absorbed and utilized by crops in the field and the total amount of water introduced by the canal head from the perspective of irrigation scientists [21–24]. The coefficient summarizes the basic data used for evaluating the efficiency and potential of agricultural irrigation, ensuring the scientific allocation of regional resources and undertaking development planning for water-saving irrigation. In addition, it provides an important basis for government departments to make macro decisions [25]. The influencing factors in water use efficiency involve many disciplines, such as climatology, hydrology, agronomy, engineering, economy, management, and institutions. The following three factors broadly summarize the key influences. (1) Natural conditions: it is recognized that complex and variable natural conditions (including climatic conditions, soil conditions, hydrological conditions, and the evolution of the irrigation area) can have a direct impact on regional water resources, and thus affect the water use efficiency of agricultural irrigation [26–29]. A typical example is that of southern China, with its humid climate and abundant rainfall, as well as lack of incentives and motivation to implement water-saving irrigation [30]. (2) The construction and management of the irrigation area: modernization of the construction and management of the irrigation area (which encompasses engineering construction, management systems and mechanisms, and the application of advanced irrigation technology) is an important component in improving irrigation water use efficiency, extreme natural disaster governance, and regional ecological sustainable development, thus providing strong support for the development of modern water-saving agriculture [25,31–35]. (3) Economic policy: an increasing number of studies have found that economic policy (including subsidy policies, water price policies, and water use restrictions) can affect the preferences and behaviors of peasant households, and provide them with incentives to use water-saving irrigation technology and to change their crop planting structure, which leads to changes in the water use efficiency of agricultural irrigation [36–38]. In addition, some scholars have found that other factors such as geographic spatial distribution [30,39], irrigation strategies and planting patterns [40–42], and crop types [9,43] have an impact on the water use efficiency of agricultural irrigation.

The diversity and complexity of influencing factors make it difficult to predict irrigation water use efficiency and, to our best knowledge, there are few predictive studies on irrigation water use efficiency. Most studies take a biological perspective and measure the water use efficiency of specific crops under different irrigation conditions [44–46] or predict the irrigation water demand for a period of time in

the future through complex multidisciplinary models [47–50]. Moreover, because the technological change of water resources utilization efficiency is often determined by exogenous forces, there is no mechanism to support the predictive theoretical derivation.

The shared socioeconomic pathways (SSPs) recently proposed by the Intergovernmental Panel on Climate Change can assist in analyzing the change and future evolution of the effective utilization coefficient of irrigation water, a key indicator of water resources technological change, from the broader perspective of climate change and the selection of socioeconomic development pathways. Thus, it can provide a new basis for the prediction of irrigation water efficiency [51]. The SSPs quantitatively describe five typical development pathways of the future social economy and distinguish coping capacity and adaptive capacity for different emission concentrations and climate change scenarios caused by the different development pathways. Thus, they can assist in predicting the change in the effective utilization coefficient of irrigation water in the future from the perspective of different social and economic development pathways [52–54]. At present, studies considering the impact of economic policy and development patterns on technological change under the influence of climate change are relatively rare, and the setting of future scenarios is subjective. There is no systematic and comparable unified standard for setting scenarios. Moreover, in terms of model settings, the classification of the simulated objects involved is oversimplified. For example, often, the simulated objects are only grouped by income level, without establishing a multi-dimensional index system to evaluate and group the simulated objects [55–57].

The Water Futures and Solutions (WFaS) extended the original SSPs framework and proposed the hydro-economic (HE) classification method, which can be combined with the scenario settings of the SSPs to group regions by their different economic and water resources conditions [58]. The extended SSPs–HE framework inputs more important features into the model, and assists in accurately setting optimal efficiency target values, convergence speeds, convergence time, and other parameters for the different regions. It also assists in simulating the curves demonstrating the technological change in water resources utilization in different regions under various scenarios, and thus lays a solid scientific foundation for predicting water resources utilization. Compared with the previous studies, which focused excessively on small-scale water-saving effects and the field of engineering, the extended SSPs–HE framework combines a broad perspective on the entire hydrographic basin with comprehensive management of water resources to evaluate different social and economic pathways selections from the angle of water use efficiency. It assists in choosing a suitable way to realize the Chinese Government's requirements for a strict water resource management system. It can also be used to analyze the influence of different socioeconomic development pathways on water use efficiency and to determine suggestions for socioeconomic improvement [47,58–60].

The scale of agricultural production in China is large and agricultural water accounts for 61.4% of total water use [61]. The study of irrigation water efficiency is of profound significance for solving the complex water resources problem in rural areas and realizing sustainable economic and social development. Since the implementation of the strict water resources management system was clearly proposed in the No. 1 document of the Communist Party of China (CPC) Central Committee in 2011, China has attached great importance to water use efficiency for agricultural irrigation, and it has been elevated to the macro and strategic level of national economic development. In the specific implementation opinions subsequently issued by the CPC, three red bottom lines on water resources management were clearly established, one of which requires China's effective utilization coefficient of irrigation water to be raised to more than 0.6 by 2030 [62]. In 2015 and 2016, the national average coefficients were 0.536 and 0.542, respectively [63,64]. To put this in context, in Israel, which has advanced water-saving irrigation technology, the effective utilization coefficient of irrigation water is above 0.9 [65].

The objective of this study is to group 31 Chinese provinces according to their different economic and water resources conditions by HE classifications. The water use scenario and parameters can be determined combined with the SSPs scenario settings and HE classification characteristics. The equation

of technological diffusion mechanism, the conditional convergence model, is the core tool to predict irrigation water use efficiency. The principle is to set the optimal efficiency target value, the convergence speed, convergence time, and other parameters, in order to simulate the curve for the effective utilization coefficient of irrigation water. The parallel aim is to understand the technological level of the water resources utilization of each province under different scenarios for specific years in the future, and find the improvement pathways for irrigation water use efficiency for specific regions.

This study attempts to solve three key scientific problems: (1) establishing an HE classification method for the evaluation of each province, (2) combining the HE classification results with the SSPs framework to set parameters for the future scenarios, and (3) establishing a conditional convergence model and using the parameters in the model for simulations.

## 2. Material and Methods

### 2.1. Study Area

At the end of 2016, China's agricultural irrigation area reached 67.13 million ha and the area of the national water-saving irrigation project reached 32.87 million ha, 9.5 of which was the low-pressure pipeline water delivery irrigation area. It accounted for 29% of the water-saving irrigation project area. The sprinkler irrigation area was 4.1 million ha, accounting for 12% of the water-saving irrigation project area. The micro irrigation area was 5.9 million ha, accounting for 18% of the water-saving irrigation project area. The highly efficient water-saving irrigation area with pipeline accounted for 59% of the water-saving irrigation area [64]. Table 1 summarizes the basic situation of agricultural irrigation in different administrative regions in 2016 [64,66].

**Table 1.** Basic situation of agricultural irrigation of different administrative regions in 2016.

| Administrative Region | Annual Average Precipitation (mm) | Total Water Resources (100 million m$^3$) | Total Agricultural Irrigation Water Consumption (TAIWC; 100 million m$^3$) | Irrigation Water Consumption per ha (IWCPH; m$^3$) | Effective Utilization Coefficients of Irrigation Water (EUCIW; Scalar) |
|---|---|---|---|---|---|
| China | 730.0 | 32466.4 | TAIWC = 3318.9 | IWCPH = 5700 | EUCIW = 0.542 |
| Beijing | 660.0 | 35.1 | TAIWC < 100 | IWCPH < 4500 | EUCIW > 0.60 |
| Tianjin | 622.1 | 18.9 | TAIWC < 100 | IWCPH < 4500 | EUCIW > 0.60 |
| Hebei | 595.9 | 208.3 | 100 < TAIWC < 200 | IWCPH < 4500 | EUCIW > 0.60 |
| Shanxi | 615.4 | 134.1 | TAIWC < 100 | IWCPH < 4500 | 0.60 > EUCIW > 0.50 |
| Inner Mongolia | 283.0 | 426.5 | 100 < TAIWC < 200 | 4500 < IWCPH < 7500 | 0.60 > EUCIW > 0.50 |
| Liaoning | 755.4 | 331.6 | TAIWC < 100 | 4500 < IWCPH < 7500 | 0.60 > EUCIW > 0.50 |
| Jilin | 731.1 | 488.8 | TAIWC < 100 | 4500 < IWCPH < 7500 | 0.60 > EUCIW > 0.50 |
| Heilongjiang | 564.2 | 843.7 | TAIWC > 200 | 4500 < IWCPH < 7500 | 0.60 > EUCIW > 0.50 |
| Shanghai | 1566.3 | 61.0 | TAIWC < 100 | 4500 < IWCPH < 7500 | EUCIW > 0.60 |
| Jiangsu | 1410.5 | 741.7 | TAIWC > 200 | 4500 < IWCPH < 7500 | EUCIW > 0.60 |
| Zhejiang | 1953.8 | 1323.3 | TAIWC < 100 | 4500 < IWCPH < 7500 | 0.60 > EUCIW > 0.50 |
| Anhui | 1612.7 | 1245.2 | 100 < TAIWC < 200 | IWCPH < 4500 | 0.60 > EUCIW > 0.50 |
| Fujian | 2503.3 | 2109.0 | TAIWC < 100 | 7500 < IWCPH < 12000 | 0.60 > EUCIW > 0.50 |
| Jiangxi | 1996.7 | 2221.1 | 100 < TAIWC < 200 | 7500 < IWCPH < 12000 | 0.50 > EUCIW > 0.40 |
| Shandong | 658.3 | 220.3 | 100 < TAIWC < 200 | IWCPH < 4500 | EUCIW > 0.60 |
| Henan | 787.1 | 337.3 | 100 < TAIWC < 200 | IWCPH < 4500 | EUCIW > 0.60 |
| Hubei | 1423.4 | 1498.0 | 100 < TAIWC < 200 | 4500 < IWCPH < 7500 | 0.60 > EUCIW > 0.50 |
| Hunan | 1668.9 | 2196.6 | 100 < TAIWC < 200 | 7500 < IWCPH < 12000 | 0.60 > EUCIW > 0.50 |
| Guangdong | 2357.6 | 2458.6 | 100 < TAIWC < 200 | 7500 < IWCPH < 12000 | 0.50 > EUCIW > 0.40 |
| Guangxi | 1631.6 | 2178.6 | 100 < TAIWC < 200 | IWCPH > 12000 | 0.50 > EUCIW > 0.40 |
| Hainan | 2341.5 | 489.9 | TAIWC < 100 | IWCPH > 12000 | 0.60 > EUCIW > 0.50 |
| Chongqing | 1236.8 | 604.9 | TAIWC < 100 | 4500 < IWCPH < 7500 | 0.50 > EUCIW > 0.40 |
| Sichuan | 921.3 | 2340.9 | 100 < TAIWC < 200 | 4500 < IWCPH < 7500 | 0.50 > EUCIW > 0.40 |
| Guizhou | 1213.7 | 1066.1 | TAIWC < 100 | 4500 < IWCPH < 7500 | 0.50 > EUCIW > 0.40 |
| Yunnan | 1295.9 | 2088.9 | TAIWC < 100 | 4500 < IWCPH < 7500 | 0.50 > EUCIW > 0.40 |
| Tibet | 611.6 | 4642.2 | TAIWC < 100 | 7500 < IWCPH < 12000 | 0.50 > EUCIW > 0.40 |
| Shaanxi | 626.2 | 271.5 | TAIWC < 100 | IWCPH < 4500 | 0.60 > EUCIW > 0.50 |
| Gansu | 290.9 | 168.4 | TAIWC < 100 | 4500 < IWCPH < 7500 | 0.60 > EUCIW > 0.50 |
| Qinghai | 304.7 | 612.7 | TAIWC < 100 | 7500 < IWCPH < 12000 | 0.50 > EUCIW > 0.40 |
| Ningxia | 301.0 | 9.6 | TAIWC < 100 | 7500 < IWCPH < 12000 | 0.60 > EUCIW > 0.50 |

Notes: The administrative regions exclude Hong Kong, Macao, and Taiwan in this study.

### 2.2. Methodology and Data for Hydro-Economic Classification

### 2.2.1. Conceptual Approach and Overview

The hydro-economic classification is a classification system for regions and watersheds that describes different conditions pertaining to water security (or its reverse, water challenges). Regions (and watersheds) are classified into a two-dimensional hydro-economic quadrant space with a compound indicator (Figure 1), which is composed of or represents the following factors [58–60]:

(1) Economic–institutional coping capacity (y-dimension): This represents the regional economic and institutional capacity to deal with water challenges. It also represents the social adaptability of a region, that is, the amount of social resources available for a region to adapt to the scarcity of natural resources. For example, Israel is short of water resources (per capita water resources are 389 m$^3$), but it can maintain a developed modern society with a per capita gross domestic product of over 10,000 U.S. dollars because of its strong economic capacity and social adaptability [67].

(2) Hydro-climatic complexity (x-dimension): This represents the magnitude/complexity of water challenges in terms of water availability and variability within and across years in a region. The hydrological system is an open, dynamic, and nonlinear complex system, which is influenced by multiple factors, such as climate, hydrometeorology, physiography, and human activity, and its long-run evolution involves both certainty and uncertainty [68]. Therefore, a region's water challenges are dynamic and variable, with the relative location of a region in the HE quadrant tending to shift over time.

(3) Hydro-economic quadrant: A two-dimensional hydro-economic quadrant space is divided into four parts. Taking the provincial scale as an example, provinces in the HE-1 quadrant (water secure, poor) are at a low-to-middle income level and face moderate hydrological challenges; provinces in the HE-2 quadrant (water secure, rich) are at a middle-to-high income level and face moderate hydrological challenges; provinces in the HE-3 quadrant (water stressed, rich) are at a middle-to-high income level and face substantial hydrological challenges; and provinces in the HE-4 quadrant (water stressed, poor) are at a low-to-middle income level and face substantial hydrological challenges.

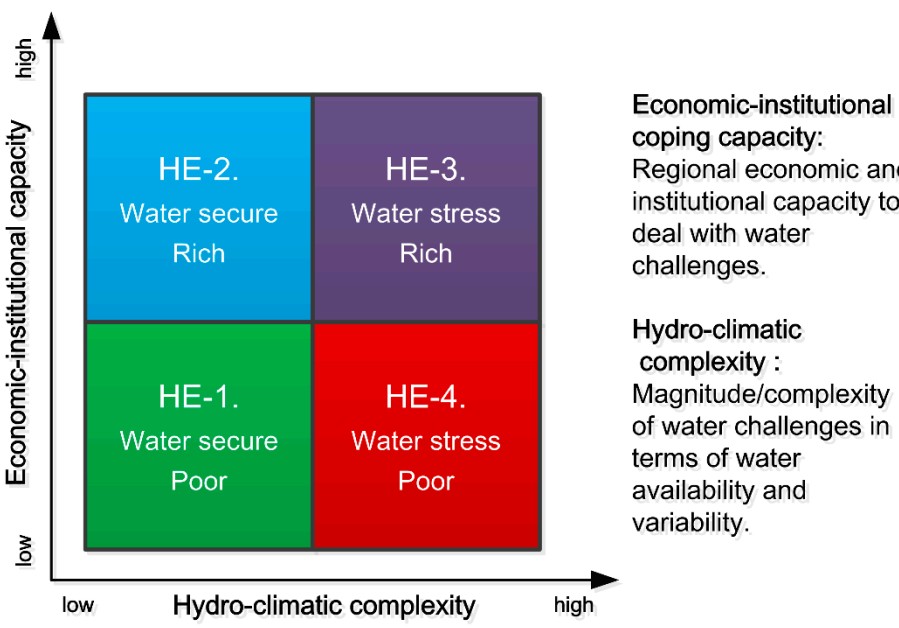

**Figure 1.** Hydro-economic (HE) quadrants for human water development challenges [58].

Each major dimension is measured using normalized indicators for classification. For the *y*-dimension, personal disposable income (PDI) is selected as a measure of economic strength and the financial resources that could be invested in risk management. The *x*-dimension is decomposed into three subindexes: total water resources per capita (TWRPC), which is a measure of water availability in each province; the ratio of annual total water withdrawal to the total water resources of each province, which is a measure of the intensity of water use (IWU); and the proportion of external water resources (from outside the regional boundaries) in the total water resources of each province, as a measure of the dependency share of external to total water resources (DS). These subindexes are standardized, weighted, and integrated into a compound index that measures the conditions and complexity of local water resources.

2.2.2. Methodology for Indicator Calculation

After selecting the relevant study scale and indicator variables for the *x*- and *y*-dimensions of the HE classification scheme, the classification process proceeds as follows [58,59]:

(1)  For each indicator variable, five classes are defined by a relevant scale (with linear or logarithmic scales determined as appropriate), and the initial index value of each class is converted into the corresponding normalized interval [0, 0.2], [0.2, 0.4], ... , [0.8, 1].

(2)  Then, we map each initial index/variable $B_i$ of i = 1,..., n, to the standardized index value $Z_i$ by the following:

    a.  determining the range of the initial index value $b_i$ for a province, b $\in$ [$B_j$, $B_{j+1}$], and

    b.  calculating the standardized index value $Z_i(b_i)$ according to the following formula:

$$Z_i(b) = Z_i(B_j) + [0, \min\left(1, \frac{b - B_j}{B_{j+1} - B_j}\right)][Z_i(B_{j+1}) - Z_i(B_j)]. \tag{1}$$

(3)  Following the World Resources Institute's aqueduct approach [69], an appropriate weight $W_i$ is set for each subindex in a nonlinear way according to the perceived importance of several classes. We selected the following weight scale:

Weight: 1 = Very Low; 2 = Low; 4 = Medium; 8 = High; 16 = Very high.

In this case, high importance (indicated by a weight of 8) was assigned to two indicators, the TWRPC and the IWU, and medium importance (a weight of 4) was assigned to the DS.

The weighted sum of the standard subindex $Z_i$ is calculated using the following formula and expressed as the compound index I :

$$I(B) = \frac{\sum_{i=1}^{n} W_i Z_i(b_i)}{\sum_{i=1}^{n} W_i}, \tag{2}$$

where B = ($b_1$,..., $b_n$) is the vector of observed (or simulated) indicators for each province.

Table 2 shows the range values of five classes used for the normalized subindex function for 2016. Data sources include the National Bureau of Statistics (PDI and population numbers), the China water resources bulletin (total water resources and total water withdrawal), and the water resources bulletins of each province (inbound water resources, outbound water resources, the south-to-north water diversion, and the Yellow River water diversion). Range values are based on the observed (or simulated) indicators and Shiklomanov's study [70]. Note that the mapping value is set to one when PDI is greater than 13,550 dollars/cap/year, which indicates very strong economic capacity, whereas a value of zero indicates very low economic capacity. To achieve the same orientation when combining the subindexes $Z_i$, the range value of TWRPC is set in reverse order (i.e., a larger TWRPC value corresponds with lower hydrological complexity and a smaller mapping value). The mapping

value is set to zero when TWRPC is greater than 20,000 m$^3$/cap/year, which indicates very low pressure on regional water sources, whereas it takes a value of one when TWRPC is less than 100 m$^3$/cap/year, which indicates very high pressure on regional water sources. The mapping value is set to one when DS is greater than 0.95, which indicates very high external dependence of regional water use, whereas it is set to zero when DS is less than 0.03, which indicates very low external dependence of regional water use.

**Table 2.** Range values of five classes used for the normalized subindex function.

| Class & Corresponding Normalized Interval | Personal Disposable Income (PDI; Dollars/Cap/Year) | Total Water Resources per Capita (TWRPC; m$^3$/Cap/Year) | Intensity of Water Use (IWU; Scalar) | Dependency Share of External to Total Water Resources (DS; Scalar) |
|---|---|---|---|---|
| CL1, [0,0.2] | 0 < PDI < 2258 | 10000 < TWRPC < 20000 | 0 < IWU < 0.05 | 0.03 < DS < 0.30 |
| CL2, [0.2,0.4] | 2258 < PDI < 3011 | 5000 < TWRPC < 10000 | 0.05 < IWU < 0.15 | 0.30 < DS < 0.45 |
| CL3, [0.4,0.6] | 3011 < PDI < 4517 | 2000 < TWRPC < 5000 | 0.15 < IWU < 0.30 | 0.45 < DS < 0.55 |
| CL4, [0.6,0.8] | 4517 < PDI < 7528 | 1000 < TWRPC < 2000 | 0.30 < IWU < 0.60 | 0.55 < DS < 0.70 |
| CL5, [0.8,1] | 7528 < PDI < 13550 | 100 < TWRPC < 1000 | 0.60 < IWU < 1.00 | 0.70 < DS < 0.95 |

*2.3. A Water Use Scenario under SSPs Framework*

The original SSPs framework does not include a water use scenario. It depicts five typical global situations with different socioeconomic conditions: SSP1 (sustainability), SSP2 (middle of the road), SSP3 (fragmentation), SSP4 (inequality), and SSP5 (conventional development) [52–54]. However, we can infer that irrigation water use must vary among these scenarios. Following Hanasaki's and Wada's studies [51,58], the clues provided by various narratives on the scenarios enable us to develop appropriate corresponding water use scenarios. Table 3 summarizes the key details of each water use scenario under the SSPs framework.

**Table 3.** Summary of the water use narrative scenarios under the shared socioeconomic pathways (SSPs) framework.

| Path-Way | Irrigated Area & Crop Intensity | Water Use Efficiency | Convergence Level & Speed | Scenario Description |
|---|---|---|---|---|
| SSP1 | Low growth | High efficiency | High level & low speed | •A long-run development concept of openness, equality, and mutual benefit. •Rapid urbanization and fast technological diffusion. •Sustainable food systems: high agricultural production efficiency and a strong preference for low-meat diets. •The whole society has a good atmosphere of energy conservation and emission reduction. |
| SSP2 | Medium growth | Medium efficiency | Medium level & very fast speed | •Moderate income growth and moderate urbanization. •Limited technological innovation and environmental protection policies and could not get rid of the middle-income trap. •Low agricultural production efficiency and a strong preference for meat consumption. •Growth in irrigation water use efficiency has slowed, and barely meets the 2030 target. |
| SSP3 | High growth | Low efficiency | Low level & fast speed | •Regional fragmentation and incompatibility. •Backward economy and ineffective environmental policies, and technology is stuck in a groove. •High population growth, low urbanization, and unscientific urban planning. •High water consumption leads to less water use for irrigation and decrease in agricultural production. |

| Path-Way | Irrigated Area & Crop Intensity | Water Use Efficiency | Convergence Level & Speed | Scenario Description |
|---|---|---|---|---|
| SSP4 | Low growth | High (developed)/ low (developing) | Medium level & medium speed | • For the regions with low hydro-climatic complexity and low income, the irrigation water use efficiency is low owing to the backward economy and limited investment in irrigation facilities. <br> • For the regions with high income, the irrigation water use efficiency could maintain a high level owing to the strong economic coping capacity. <br> • For the regions with the dual pressure of backward economy and hydro-climatic complexity, the irrigation water use efficiency is in a low level. <br> • Technologies diffuse across the regions with different economic development level. |
| SSP5 | High growth | High efficiency | High level & very low speed | • A conventional fossil-fueled pathway with the rapid capital accumulation and massive greenhouse gas emissions. <br> • Strong technological progress in the agricultural sector. <br> • Highly managed and resource intensive agro-ecosystems and water systems. |

## 2.4. Conditional Convergence Model

The conditional convergence for predicting water use efficiency is based on the following three assumptions:

(1) The efficiency level in a specific region gradually converges to the optimum.
(2) There are two modes of technological development, namely technological transmission in advanced regions and technological catch-up in backward regions.
(3) In the same period of time, the speed of improvement in the region with advanced technology is slower than that in the region with backward technology.

The specific calculation formula is as follows [71–73]:

$$E_r(t) = E_A^L + \left(E_r(0) - E_A^L\right) \cdot e^{-\Delta t \beta(r)}, \tag{3}$$

where $E_r(t)$ (scalar) represents water use efficiency in convergence time $t$ (years), $E_A^L$ (scalar) represents water use efficiency for medium- to long-term targets, $E_r(0)$ (scalar) represents initial water use efficiency in a region, and $\beta(r)$ (scalar) represents the convergence control parameters in a specific region.

## 3. Scenario Determination and HE Evaluation

### 3.1. Scenario and Parameter Setting Under the SSPs–HE Framework

Combining water use narrative scenarios under the SSPs framework (the five specific pathways) and various HE classifications enables us to determine different convergence targets, convergence rates, and other parameters for provinces in different quadrants of the hydro-economic quadrant space.

The 2030 national target for the coefficient of effective utilization of irrigation water is established in *Views on the implementation of the strictest water resources management system*. The 2030 target value of each province can be converted by the 2015 coefficient of each province [63], which can be taken as a benchmark of the convergence target for further simulation.

Table 4 provides a qualitative description of irrigation water use efficiency under the SSPs. It describes the level of irrigation water use efficiency in various quadrant provinces under the five different socioeconomic pathways defined above. For example, SSP2 is a medium scenario in every HE classification; SSP5 presents high efficiency in every HE classification. SSP2 is also more pessimistic about the target of convergence than SSP5 and has a shorter convergence time than SSP5. Table 5 shows the quantitative transformation of the convergence parameters used for further simulations. It describes the convergence targets (multiples of the benchmark target) and the convergence times (years) in various quadrant provinces under different socioeconomic pathways.

**Table 4.** Qualitative description of irrigation water use efficiency under SSPs.

| Pathway | HE-1 | HE-2 | HE-3 | HE-4 |
|---------|------|------|------|------|
| SSP1 | High | Medium-high | Medium-high | High |
| SSP2 | Medium | Medium | Medium | Medium |
| SSP3 | Medium-low | Low | Medium | Medium-low |
| SSP4 | Low | Medium-high | Medium-high | Low |
| SSP5 | High | High | High | High |

To our best knowledge, there have been few studies on the impact of economic and social development modes on technological change. Thus, setting the future scenario assumptions could be subjective because of the lack of a systematic and comparable unified scenario-setting standard. Thus, the relevant parameter settings in this study are largely based on the classical theory of technological change and the previous experience of multifactor and total factor productivity in Organization for Economic Co-operation and Development (OECD) and major non-OECD economies [71–74].

*3.2. HE Classification Evaluation*

Following the calculation process for the HE classification, we obtain the results in Tables 5 and 6, which show the mapping values of economic–institutional capacity (the *y*-dimension) and hydro-climatic complexity (the *x*-dimension), respectively, for 31 Chinese provinces (excluding Hong Kong, Macao, and Taiwan) for 2016. Then, the quadrants scatter diagram and the national distribution of the HE classification of the 31 Chinese provinces for 2016 are depicted in Figure 2.

**Table 5.** Quantitative transformation of convergence parameters for simulation.

| Pathway | HE-1 | | HE-2 | | HE-3 | | HE-4 | |
|---|---|---|---|---|---|---|---|---|
| | Convergence Target (Multiple of Benchmark Target) | Convergence Time (Years) | Convergence Target (Multiple of Benchmark Target) | Convergence Time (Years) | Convergence Target (Multiple of Benchmark Target) | Convergence Time (Years) | Convergence Target (Multiple of Benchmark Target) | Convergence Time (Years) |
| SSP1 | 1.1 | 100 | 1.1 | 50 | 1.1 | 50 | 1.1 | 100 |
| SSP2 | 1.0 | 15 | 1.0 | 15 | 1.0 | 15 | 1.0 | 15 |
| SSP3 | 0.9 | 30 | 0.9 | 30 | 1.0 | 15 | 0.9 | 50 |
| SSP4 | 1.0 | 50 | 1.1 | 50 | 1.1 | 50 | 1.0 | 50 |
| SSP5 | 1.1 | 100 | 1.1 | 100 | 1.1 | 100 | 1.1 | 100 |

**Table 6.** Mapping results of the economic–institutional capacity of 31 provinces in 2016.

| Province | Personal Disposable Income (PDI) in 2016 (Dollars/Cap/Year) | The Mapping Value of the Y-Dimension (Economic–Institutional Capacity) | Province | Personal Disposable Income (PDI) in 2016 (Dollars/Cap/Year) | The Mapping Value of the Y-Dimension (Economic–Institutional Capacity) |
|---|---|---|---|---|---|
| Anhui | 3010.72 | 0.400 | Liaoning | 3920.28 | 0.521 |
| Beijing | 7908.46 | 0.813 | Inner Mongolia | 3632.27 | 0.483 |
| Fujian | 4156.38 | 0.552 | Ningxia | 2835.20 | 0.353 |
| Gansu | 2208.62 | 0.196 | Qinghai | 2604.78 | 0.292 |
| Guangdong | 4561.04 | 0.603 | Shandong | 3716.37 | 0.494 |
| Guangxi | 2755.83 | 0.332 | Shanxi | 2867.81 | 0.362 |
| Guizhou | 2276.49 | 0.205 | Shaanxi | 2841.45 | 0.355 |
| Hainan | 3109.38 | 0.413 | Shanghai | 8175.68 | 0.822 |
| Hebei | 2969.67 | 0.389 | Sichuan | 2831.59 | 0.352 |
| Henan | 2776.61 | 0.338 | Tianjin | 5129.92 | 0.641 |
| Heilongjiang | 2986.69 | 0.394 | Tibet | 2053.39 | 0.182 |
| Hubei | 3279.98 | 0.436 | Xinjiang | 2763.30 | 0.334 |
| Hunan | 3178.84 | 0.422 | Yunnan | 2517.19 | 0.269 |
| Jilin | 3006.04 | 0.399 | Zhejiang | 5800.55 | 0.685 |
| Jiangsu | 4828.16 | 0.621 | Chongqing | 3317.25 | 0.441 |
| Jiangxi | 3027.50 | 0.402 | - | - | - |

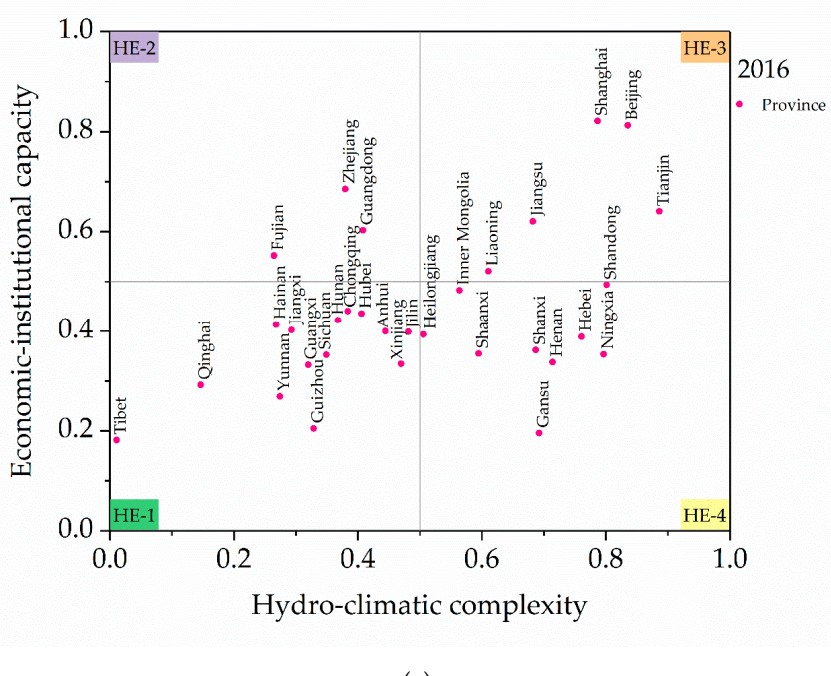

(**a**)

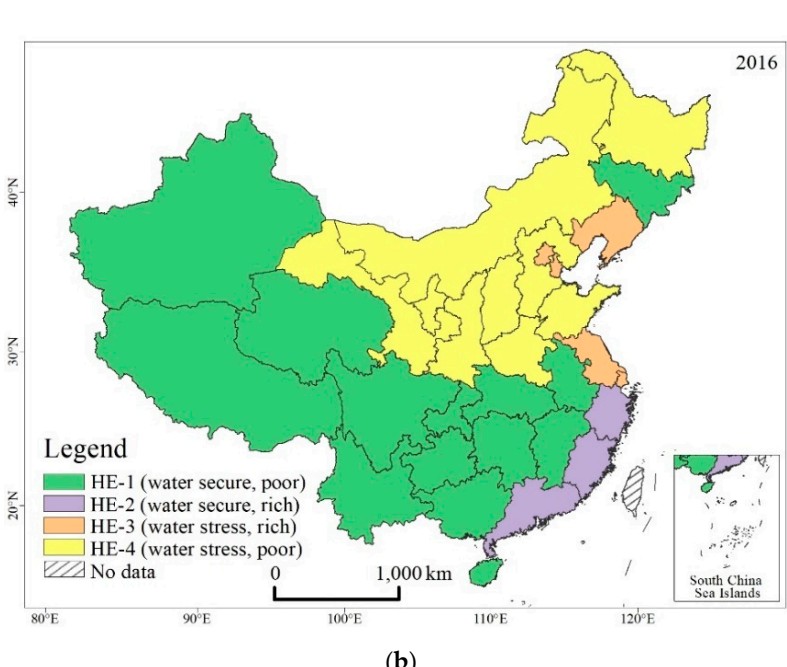

(**b**)

**Figure 2.** (**a**) HE classification quadrants of 31 Chinese provinces for 2016; (**b**) national distribution of HE classification for 31 Chinese provinces for 2016.

In Table 6, PDI (dollars/cap/year) ranges from 2053.39 for Tibet to 8175.68 for Shanghai. The mapping values for Tibet and Shanghai are 0.182 and 0.822, respectively. In Table 7, the observed values of the subindexes (TWRPC, IWU, and DS) are converted into the corresponding mapping values and the compound weighted index I ranges from zero for Tibet to 0.886 for Tianjin.

Figure 2a shows the relative location of all provinces in the HE quadrant space. A total of 14 provinces are in the HE-1 quadrant (Anhui, Guangxi, Guizhou, Hainan, Hubei, Hunan, Jilin, Jiangxi, Qinghai, Sichuan, Tibet, Xinjiang, Yunnan, and Chongqing); three provinces are in the HE-2 quadrant (Fujian, Guangdong, and Zhejiang); five provinces are in the HE-3 quadrant (Beijing, Jiangsu,

Liaoning, Shanghai, and Tianjin); and nine provinces are in the HE-4 quadrant (Gansu, Hebei, Henan, Heilongjiang, Inner Mongolia, Ningxia, Shandong, Shanxi, and Shaanxi). Figure 2b shows the distribution of these provinces. Provinces in southwestern China generally face low hydrological challenges, whereas most provinces in northeastern China face substantial hydrological challenges. A few provinces in the coastal area of eastern China have strong economic capacity, whereas most provinces inland have comparatively low income levels. These classification results are consistent with an intuitive assessment. Note that, for simplicity, this study follows the WFaS analysis [58] and retains the classification results only in 2016.

**Table 7.** Weighted and mapping results for the hydro-climatic complexity of 31 provinces in 2016.

| Subindex of X-Dimension | TWRPC | IWU | DS | The Compound Index (Hydro-Climatic Complexity) | Subindex of X-Dimension | TWRPC | IWU | DS | The Compound Index (Hydro-Climatic Complexity) |
|---|---|---|---|---|---|---|---|---|---|
| Weight | | | | | Weight | | | | |
| Province | 8 | 8 | 4 | I | Province | 8 | 8 | 4 | I |
| Tianjin | 0.995 | 1.000 | 0.440 | 0.886 | Anhui | 0.599 | 0.511 | 0.000 | 0.444 |
| Beijing | 0.986 | 1.000 | 0.204 | 0.835 | Guangdong | 0.584 | 0.436 | 0.000 | 0.408 |
| Shandong | 0.973 | 0.986 | 0.088 | 0.801 | Hubei | 0.564 | 0.451 | 0.000 | 0.406 |
| Ningxia | 0.991 | 1.000 | 0.000 | 0.796 | Chongqing | 0.603 | 0.356 | 0.000 | 0.384 |
| Shanghai | 0.966 | 1.000 | 0.000 | 0.786 | Zhejiang | 0.576 | 0.374 | 0.000 | 0.380 |
| Hebei | 0.960 | 0.938 | 0.006 | 0.761 | Hunan | 0.519 | 0.401 | 0.000 | 0.368 |
| Henan | 0.944 | 0.837 | 0.007 | 0.714 | Sichuan | 0.544 | 0.328 | 0.000 | 0.349 |
| Gansu | 0.879 | 0.852 | 0.000 | 0.692 | Guizhou | 0.533 | 0.288 | 0.000 | 0.329 |
| Shanxi | 0.941 | 0.775 | 0.000 | 0.687 | Guangxi | 0.433 | 0.367 | 0.000 | 0.320 |
| Jiangsu | 0.816 | 0.889 | 0.000 | 0.682 | Jiangxi | 0.411 | 0.321 | 0.000 | 0.293 |
| Liaoning | 0.854 | 0.672 | 0.000 | 0.610 | Yunnan | 0.441 | 0.244 | 0.000 | 0.274 |
| Shaanxi | 0.864 | 0.623 | 0.000 | 0.595 | Hainan | 0.386 | 0.284 | 0.000 | 0.268 |
| Inner Mongolia | 0.662 | 0.697 | 0.100 | 0.564 | Fujian | 0.382 | 0.279 | 0.000 | 0.265 |
| Heilongjiang | 0.585 | 0.679 | 0.000 | 0.506 | Qinghai | 0.193 | 0.172 | 0.000 | 0.146 |
| Jilin | 0.642 | 0.561 | 0.000 | 0.481 | Tibet | 0.000 | 0.027 | 0.000 | 0.011 |
| Xinjiang | 0.429 | 0.745 | 0.000 | 0.470 | - | - | - | - | - |

## 4. Simulation and Results Analysis

### 4.1. Prediction of the Irrigation Water Use Efficiency of Each Province

On the basis of the HE classification results for each province and the description of irrigation water use scenarios under different socioeconomic pathways, we can further predict the effective utilization coefficients of irrigation water in each province under different SSPs, as shown in Figure 3.

Taking an overall view of the five development pathways, the effective utilization coefficients of irrigation water in half of the provinces converge to 0.6 under SSP1, SSP2, and SSP5 by 2030, whereas only nine provinces reach 0.6 under SSP3 and 11 provinces do under SSP4. The HE-3 class has the highest proportion (100%) of provinces that can achieve the irrigation water use efficiency target under every development pathway in 2030, followed by the HE-4 class (64%), and then the HE-2 class (53%). The HE-1 class has the lowest proportion (9%) of provinces successfully reaching the 0.6 target. Provinces with severe hydrological conditions generally have higher water use efficiency than do provinces with low hydrological challenges.

Comparing the simulation values in 2016 and 2030, HE-3 provinces present the largest improvement in efficiency. All HE-3 provinces have a coefficient of more than 0.6 under each pathway in 2030. Indeed, with the except of Liaoning, the coefficients of these provinces are above 0.7, with Shanghai close to 0.9. Moreover, the differences in the coefficients between the five pathways are relatively small for the HE-3 provinces, with a standard deviation below 0.02.

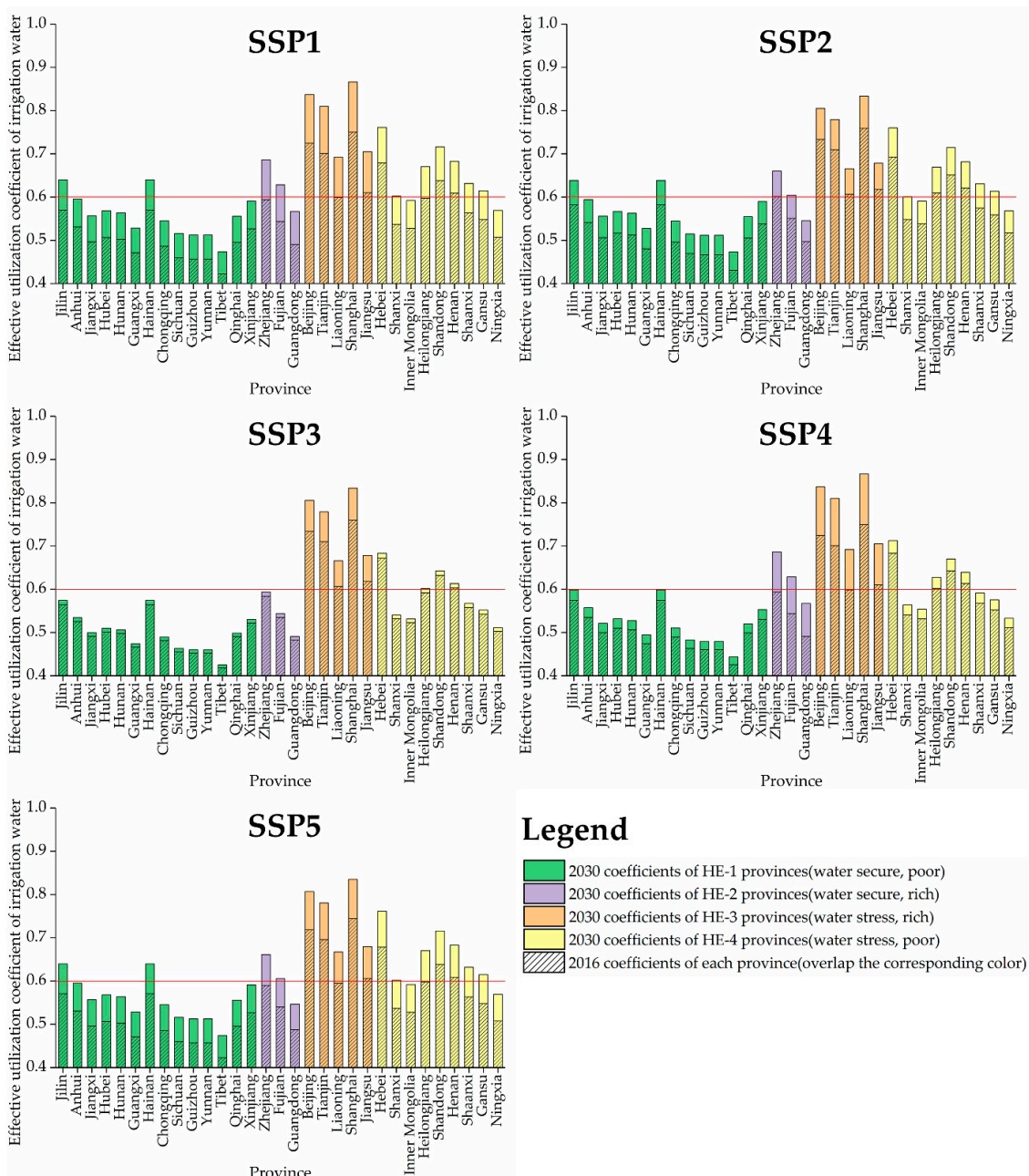

**Figure 3.** Effective utilization coefficients of irrigation water in each province under different shared socioeconomic pathways (SSPs) in 2016 and 2030.

For HE-2 provinces, Zhejiang has the highest effective irrigation water use coefficient, followed by Fujian and Guangdong. Under SSP3, the coefficients of these three provinces are all lower than the target of 0.6 in 2030. However, under the other pathways, Zhejiang and Fujian do achieve coefficients higher than 0.6, although Guangdong still fails to reach 0.6 by 2030.

For HE-1 provinces, only Jilin and Hainan reach the 0.6 target under SSP1, SSP2, and SSP5. The other provinces in the HE-1 class have relatively low efficiency compared with all the HE provinces because abundant resources mean there is no pressure to reduce water use and there is limited investment in water-saving facilities.

Provinces in HE-4 are facing large uncertainties regarding water use efficiency in the future owing to their backward economies and strong pressure on scarce water resources. HE-4 provinces have the largest fluctuations in water use efficiency under the different pathways of all four HE classifications,

with a standard deviation of more than 0.02 in 2030. The standard deviation in Hebei and Shandong is even higher, at 0.03. The effective water use coefficient of most provinces in the HE-4 quadrant would be above 0.6 in 2030 under SSP1, SSP2, SSP4, and SSP5. Only Ningxia and Inner Mongolia would be below 0.6, but both would be very close to reaching this target.

### 4.2. Analysis for Typical HE Provinces

On the basis of the relative location of the provinces in HE quadrant space, Hubei, Guangdong, Jiangsu, and Gansu are selected as representatives of all provinces in HE-1, HE-2, HE-3, and HE-4 quadrants, respectively, and used to illustrate the convergence of the effective utilization coefficients of irrigation water in these four classes under different pathways from 2016 to 2030 (Figure 4).

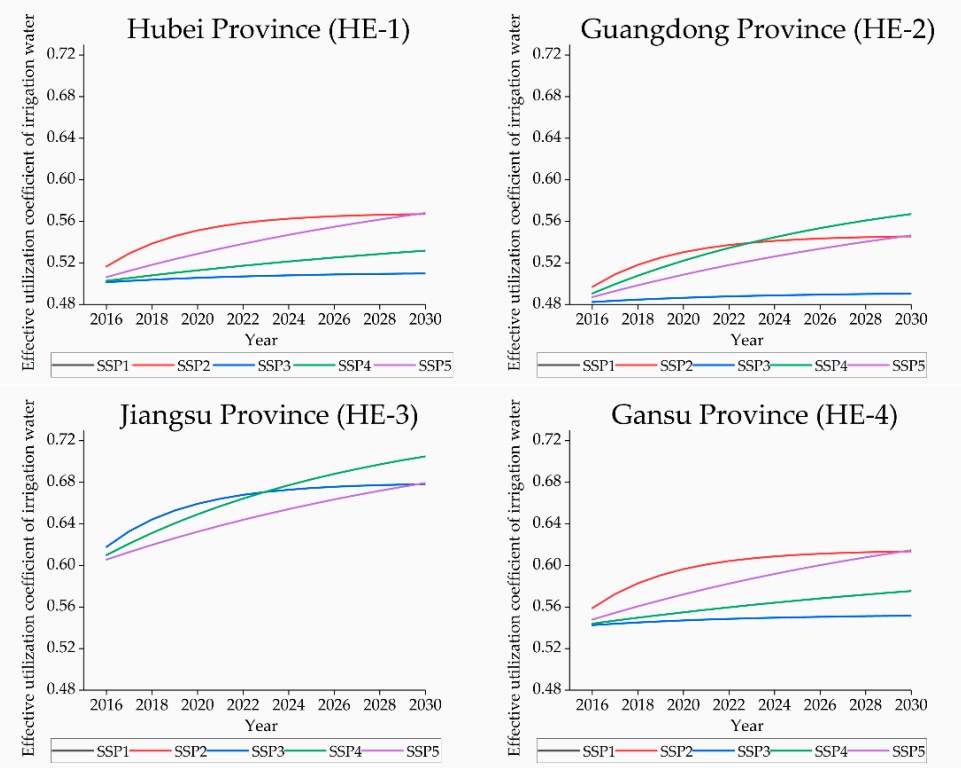

**Figure 4.** Convergence of effective utilization coefficients of irrigation water of four typical HE provinces under different SSPs, 2016–2030.

Jiangsu, a province with high pressure on water use and a strong economy, has a high effective utilization coefficient of irrigation water under the five pathways, and easily reaches the national standard of 0.6 by 2030. Jiangsu irrigation water use efficiency is highest under SSP1 and SSP4 (as the two curves coincide), and Jiangsu would remain in the efficient growth period with adequate potential for technological change. Its irrigation water use efficiency under SSP5 is the third-best case, with the coefficient reaching 0.68 in 2030, but with relatively narrow scope remaining for technological change improvements compared with SSP1 and SSP4. Under SSP3, the efficiency of irrigation water converges quickly but essentially remains unchanged after reaching 0.68, and there is no room for further technological improvement.

Gansu, which is under strong pressure to reduce water use and has a weak economy, has a generally lower irrigation water use efficiency compared with that of Jiangsu. Owing to adequate capital and open channels for technological transmission, the irrigation water use efficiency is at a relatively high level under SSP1 and SSP5 (the two curves coincide). Not only would the irrigation water use efficiency target be reached by 2030, but there would remain room for improvement in the future, although the

improvement rate would be lower than that in Jiangsu. Under SSP2, because recent policies have placed strong emphasis on water saving, the irrigation water use efficiency would improve rapidly and then converge to 0.6. However, it would then remain unchanged because of a lack of financial support and slow technological diffusion. For the highly unbalanced SSP4 scenario, backward provinces such as Gansu are at a disadvantage because they lack capital and talent. The demonstration effect of advanced provinces and the diffusion effect of advanced technologies means that the irrigation water use efficiency can be improved slowly through partial catch-up. The national target will not be reached by 2030, however, and irrigation water use efficiency in Gansu remains basically stagnant under SSP3.

Hubei is in the third place among the four provinces in terms of the overall irrigation water use efficiency situation, performing below Jiangsu and Gansu, but better than Guangdong. Because it has abundant water resources, Hubei lacks the motivation to improve water use efficiency and it will not meet the national target by 2030 under any pathways. It is similar to Gansu province in regard to the low coefficient for the effective utilization of irrigation water and other conditions.

Among the four provinces with different HE classifications, Guangdong in the HE-2 class has the lowest irrigation water use efficiency because of the absence of pressure on its water resources, which limits improvements in water use efficiency. The speed of improvement is highest under SSP1 and SSP4 (the two curves coincide), but even under these pathways, Guangdong is unable to meet the national target by 2030. Under SSP5, the next best scenario, which is based on fossil fuels, it fails to reach the convergence state by 2030. Guangdong reaches convergence soonest under SSP2 and its coefficient value (0.55) is close to SSP5 by 2030. Under SSP3, the province is in a stable state in which irrigation water use efficiency is stagnant and always lower than 0.5 up to 2030.

## 5. Conclusions and Suggestions

This study uses a conditional convergence model for predicting technological change combined with the SSPs scenario settings and HE classification to group 31 Chinese provinces by their different economic and water resources conditions. On this basis, it presents the results from a new extended SSPs–HE framework for predicting the change in irrigation water use efficiency of 31 Chinese provinces by 2030. The conclusions are as follows.

The effective utilization coefficients of irrigation water in half of the provinces converge to 0.6 under SSP1, SSP2, and SSP5 by 2030, whereas SSP3 and SSP4 are generally inefficient development pathways. The HE-3 class has the highest proportion (100%) of provinces that can achieve the irrigation water use efficiency target under every development pathway in 2030, followed by the HE-4 class (64%), and then the HE-2 class (53%). The HE-1 class has the lowest proportion (9%) of successful provinces.

Provinces with severe hydrological conditions generally have higher water use efficiency than do provinces with low hydrological challenges. Substantial regional hydrological challenges are the most important incentive or internal driving force to improve water use efficiency. In addition, HE-3 provinces present the largest improvement in irrigation water use efficiency, reflecting the great importance of economic capacity in improving water use efficiency.

The curves for the effective utilization coefficients of irrigation water have different trajectories in the different scenarios. Coefficient curves increase rapidly in various provinces under SSP1 and SSP5, but these pathways do not result in provinces reaching the convergence state by 2030 and room for further improvement remains. Coefficient curves have the fastest convergence rate under SSP2 among all the pathways, but this pathway lacks the potential for further development and provinces remains stagnant once they converge to a certain level. The coefficient curves present an inefficient situation under SSP3, in which the irrigation water use efficiency is at a low level and does not significantly improve for a long time. Under SSP4, the coefficient curves reflect a highly imbalanced situation with coefficient curve trajectories depending on the economic strength of the provinces. The coefficient curves of the provinces with strong economic capacity rise quickly, whereas those of the provinces with weak economic capacity rise slowly (i.e., the strong get stronger, but the weak become weaker).

On the basis of the analysis of the results and conclusions above, we put forward the following suggestions for improving the irrigation water use efficiency of each province.

A water-saving development pathway for specific regions should be selected in line with the local conditions. HE-3 provinces have both the motivation and the economic capacity for water-saving actions, with strong endogenous water-saving powers making SSP1 a suitable pathway for HE-3 provinces. HE-4 provinces are facing the dual pressures of capital shortages and severe hydrological challenges, and they require policy and financial support from the central government, including access to the limited funds designated for water-saving projects. SSP2 is likely to be the most suitable pathway for HE-4 provinces because of its cost-effectiveness. For HE-1 provinces with abundant water resources but weak economies, efforts should be made to enhance economic capacity under SSP1 or SSP5, with efficiency improved slowly, but continuously over the long term. For HE-2 provinces with strong economic capacity and water security, a harmonious relationship between maintaining human living standards and environmental water resources should be the goal of future development, which aligns with the development concept of SSP1.

Regional rivalry and fragmentation are not wise development choices. It is important to increase connectivity and openness among the regions and narrow the technological and income gaps. China is a vast country with complex national conditions and its development is inadequate and unbalanced, as evidenced by the great variation in hydro-economic conditions among regions and provinces. Although the hydro-economic conditions of backward regions cannot be changed in the short term, these regions can benefit from technological diffusion and spillovers, which requires overall irrigation water use planning at the national level and an open, inclusive, and shared development concept among all regions to change the situation of imbalance and even regional rivalry.

It is always good to be prepared, even in a province in which pressure on water resources is low, by improving the management of irrigation and the application and popularization of related technologies. The low-pressure pipeline water delivery irrigation is likely to be the most suitable technology for HE-1 provinces (low economic capacity, low hydrological challenges) because of its cost-effectiveness. For a province in which economic capacity is strong, more advanced technologies can be considered, such as the sprinkler irrigation and the micro irrigation. Considering that substantial regional hydrological challenges are the most important incentive or internal driving force to improve water use efficiency and water resources is so valuable in arid and semi-arid regions in northern China, efforts should be made continuously to develop highly efficient water-saving irrigation, such as the micro irrigation, despite the fact that some provinces have a weak economy. In addition to increasing capital investment in water-saving technology, it is very important to strengthen the belief in technological innovation and green development [75] and to build a policy and social environment that encourages technological innovation and water and energy conservation.

**Author Contributions:** Conceptualization, A.G. and F.Z.; Data curation, D.J., F.Z., and X.D.; Formal analysis, D.J., F.Z., and X.D.; Funding acquisition, F.Z.; Methodology, F.Z. and X.D.; Project administration, F.Z.; Resources, X.S. and C.H.; Supervision, A.G., F.Z., and C.H.; Visualization, D.J., X.D., and Y.Z.; Writing—Original draft, D.J. and F.Z.; Writing—Review & editing, D.J., F.Z., X.S., and Q.C.

**Funding:** This research was supported by the Strategic Priority Research Program of the Chinese Academy of Sciences [grant numbers XDA19040500 and XDA20100104]; the National Natural Science Foundation of China [grant numbers 41571516 and 41690144]; and the Fundamental Research Funds for the Central Universities (grant number 2019jbkyjd013).

**Conflicts of Interest:** The authors declare no conflict of interest.

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
