# Peer review of "Prediction of Technological Change under Shared Socioeconomic Pathways and Regional Differences: A Case Study of Irrigation Water Use Efficiency Changes in Chinese Provinces"

_sustainability, doi:10.3390/su11247103_

Round 1
Reviewer 1 Report
Dear Authors The work presented under the title "Prediction of technological change under Shared Socioeconomic Pathways and regional differences: A case study of irrigation water use efficiency changes in Chinese provinces", seems to me of great interest and that it can be a good contribution in this field of study. The document presents a qualitative and quantitative methodology to predict the level of efficiency in water use in China at the regional level. The results allow establishing a series of recipes to improve in this regard.
However, in my opinion, some aspects should be improved.
1. The introduction seems to me that it does not faithfully characterize the context of the work theme. In my opinion, aspects of sustainability and efficiency of water use in agriculture should be deepened, including references to the latest global review work such as:
Kang and Kang 2019. On the use of alternative water use efficiency parameters in dryland ecosystems: a review
Mutema et al. 2019. Factors affecting crop water use efficiency: A worldwide meta-analysis
Juan F. Velasco-Muñoz et al. 2019. Sustainable Irrigation in Agriculture: An Analysis of Global Research.
Juan F. Velasco-Muñoz et al. 2019. Rainwater Harvesting for Agricultural Irrigation: An Analysis of Global Research
Juan F. Velasco-Muñoz et al. 2018. Advances in Water Use Efficiency in Agriculture: A Bibliometric Analysis.
In addition, in the introduction concepts are named as conditional convergence model, which I think should be defined in that section. The concept of water use efficiency is defined, but I would like you to specify the perspective of that definition (agronomic, physiological), which implies a great nuance.
Finally, the exposition of the three key scientific problems is very clear. However, the presentation of the objective seems somewhat more confusing. I suggest restructuring to make it clearer.
2. With regard to the methodology section, it seems to me that the structure to follow should be the same as in the presentation of the key problems. I see an inconsistency in that aspect. In addition, I would like to explain a little better what has been the process to establish the narrative scenarios. In the current writing it seems to me that it has been a bit arbitrary. I imagine that there will be some consultation with experts, or some other methodological process.
3. I miss a general description of the study area. In my opinion it would be useful to have a table with the extension of the agricultural area, the types of crops, the irrigation systems used, etc. in each of the regions.
4. Finally, it seems to me that the conclusions are too general for such a detailed study at the regional level. Would it be possible to include more concrete measures?
Author Response
Response to Reviewer 1 Comments
Point 1: The introduction seems to me that it does not faithfully characterize the context of the work theme. In my opinion, aspects of sustainability and efficiency of water use in agriculture should be deepened, including references to the latest global review work such as:
Kang and Kang 2019. On the use of alternative water use efficiency parameters in dryland ecosystems: a review
Mutema et al. 2019. Factors affecting crop water use efficiency: A worldwide meta-analysis.
Juan F. Velasco-Muñoz et al. 2019. Sustainable Irrigation in Agriculture: An Analysis of Global Research.
Juan F. Velasco-Muñoz et al. 2019. Rainwater Harvesting for Agricultural Irrigation: An Analysis of Global Research
Juan F. Velasco-Muñoz et al. 2018. Advances in Water Use Efficiency in Agriculture: A Bibliometric Analysis.
In addition, in the introduction concepts are named as conditional convergence model, which I think should be defined in that section. The concept of water use efficiency is defined, but I would like you to specify the perspective of that definition (agronomic, physiological), which implies a great nuance.
Finally, the exposition of the three key scientific problems is very clear. However, the presentation of the objective seems somewhat more confusing. I suggest restructuring to make it clearer.
Response 1: Thanks a lot for the comments. We have supplemented the context of the work theme and all the references you recommended to us have been cited. For example:
……Technological change in agriculture irrigation water use efficiency is of fundamental significance for solving water scarcity and increasing crop productivity, and achieving highly efficient irrigation is crucial to balancing water resources input and sustainable agricultural economic growth [7-9]..….
References:
…….
Velasco-Muñoz, F. J.; Aznar-Sánchez, A. J.; Batlles-delaFuente, A.; Fidelibus, D. M., Sustainable Irrigation in Agriculture: An Analysis of Global Research. Water 2019, 11, (9). Velasco Muñoz, J.; Aznar-Sánchez, J. A.; Batlles de la Fuente, A.; Fidelibus, M., Rainwater Harvesting for Agricultural Irrigation: An Analysis of Global Research. Water 2019, 11, 1320. Mbava, N.; Mutema, M.; Zengeni, R.; Shimelis, H.; Chaplot, V., Factors affecting crop water use efficiency: A worldwide meta-analysis. Agricultural Water Management 2020, 228, 105878.…….
The concept of conditional convergence model has been defined in the introduction, which refers to an exponential model that reflects the long-term changes in technological efficiency of economies with similar structural characteristics. The concept of water use efficiency has also been specifically defined. Following Velasco-Muñoz’s and Kang’s studies, the concept of water use efficiency in this study refers to the ratio of the amount of water that can be absorbed and utilized by crops in the field and the total amount of water introduced by the canal head from the perspective of irrigation scientists.
References:
…….
Kang, W.; Kang, S., On the use of alternative water use efficiency parameters in dryland ecosystems: a review. Journal of Ecology and Environment 2019, 43, (24). Velasco-Muñoz, F. J.; Aznar-Sánchez, A. J.; Belmonte-Ureña, J. L.; López-Serrano, J. M., Advances in Water Use Efficiency in Agriculture: A Bibliometric Analysis. Water 2018, 10, (4).…….
The objective of this study has been restructured to make it clearer. For example:
The objective of this study is to group 31 Chinese provinces according to their different economic and water resources conditions by HE classifications. The water use scenario and parameters can be determined combined with the SSPs scenario settings and HE classification characteristics. The equation of technological diffusion mechanism, the conditional convergence model, is the core tool to predict irrigation water use efficiency.
Point 2: With regard to the methodology section, it seems to me that the structure to follow should be the same as in the presentation of the key problems. I see an inconsistency in that aspect. In addition, I would like to explain a little better what has been the process to establish the narrative scenarios. In the current writing it seems to me that it has been a bit arbitrary. I imagine that there will be some consultation with experts, or some other methodological process.
Response 2: Thanks a lot for the comments. We have added a new section 2.1. “Study area” in section 2. The section “A water use scenario under SSPs framework” has been moved to section 2.3. to make it more consistent with the presentation of the key problems. The establishment of the water use scenario under the SSPs framework model mainly refers to Hanasaki’s and Wada’s study and the clues provided by various narratives on the scenarios.
Reference:
Hanasaki, N.; Fujimori, S.; Yamamoto, T.; Yoshikawa, S.; Masaki, Y.; Hijioka, Y.; Kainuma, M.; Kanamori, Y.; Masui, T.; Takahashi, K.; Kanae, S., A global water scarcity assessment under Shared Socio-economic Pathways – Part 1: Water use. Hydrol. Earth Syst. Sci. 2013, 17, (7), 2375-2391.
Wada, Y.; Flörke, M.; Hanasaki, N.; Eisner, S.; Fischer, G.; Tramberend, S.; Satoh, Y.; van Vliet, M.; Yillia, P. T.; Ringler, C.; Burek, P.; Wiberg, D., Modeling global water use for the 21st century: Water Futures and Solutions (WFaS) initiative and its approaches. Geoscientific Model Development 2016, 9, 175-222.
Point 3: I miss a general description of the study area. In my opinion it would be useful to have a table with the extension of the agricultural area, the types of crops, the irrigation systems used, etc. in each of the regions.
Response 3: Thanks a lot for the comments. We have added a new section 2.1. “Study area” in section 2. Considering the data availability and thematic relevance, we include annual average precipitation, total water resources, total agricultural irrigation water consumption, irrigation water consumption per ha, effective utilization coefficients of irrigation water in a newly established table 1. For example:
2.1. Study area
At the end of 2016, China's agricultural irrigation area reached 67.13 million ha and the area of the national water-saving irrigation project reached 32.87 million ha, 9.5 of which was the low-pressure pipeline water delivery irrigation area. It accounted for 29% of the water-saving irrigation project area. The sprinkler irrigation area was 4.1 million ha, accounting for 12% of the water-saving irrigation project area. The micro irrigation area was 5.9 million ha, accounting for 18% of the water-saving irrigation project area. The highly efficient water-saving irrigation area with pipeline accounted for 59% of the water-saving irrigation area[64]. Table 1 summarizes the basic situation of agricultural irrigation in different administrative regions in 2016 [64, 66].
Table 1. Basic situation of agricultural irrigation of different administrative regions in 2016.
Administrative region |
Annual average precipitation(mm) |
Total water resources(100million m3) |
Total agricultural irrigation water consumption (TAIWC; 100million m3) |
Irrigation water consumption per ha (IWCPH; m3) |
Effective utilization coefficients of irrigation water (EUCIW; scalar) |
China |
730.0 |
32466.4 |
TAIWC=3318.9 |
IWCPH=5700 |
EUCIW=0.542 |
Beijing |
660.0 |
35.1 |
TAIWC<100 |
IWCPH<4500 |
EUCIW>0.60 |
Tianjin |
622.1 |
18.9 |
TAIWC<100 |
IWCPH<4500 |
EUCIW>0.60 |
Hebei |
595.9 |
208.3 |
100<TAIWC<200 |
IWCPH<4500 |
EUCIW>0.60 |
Shanxi |
615.4 |
134.1 |
TAIWC<100 |
IWCPH<4500 |
0.60>EUCIW>0.50 |
Inner Mongolia |
283.0 |
426.5 |
100<TAIWC<200 |
4500<IWCPH<7500 |
0.60>EUCIW>0.50 |
Liaoning |
755.4 |
331.6 |
TAIWC<100 |
4500<IWCPH<7500 |
0.60>EUCIW>0.50 |
Jilin |
731.1 |
488.8 |
TAIWC<100 |
4500<IWCPH<7500 |
0.60>EUCIW>0.50 |
Heilongjiang |
564.2 |
843.7 |
TAIWC>200 |
4500<IWCPH<7500 |
0.60>EUCIW>0.50 |
Shanghai |
1566.3 |
61.0 |
TAIWC<100 |
4500<IWCPH<7500 |
EUCIW>0.60 |
Jiangsu |
1410.5 |
741.7 |
TAIWC>200 |
4500<IWCPH<7500 |
EUCIW>0.60 |
Zhejiang |
1953.8 |
1323.3 |
TAIWC<100 |
4500<IWCPH<7500 |
0.60>EUCIW>0.50 |
Anhui |
1612.7 |
1245.2 |
100<TAIWC<200 |
IWCPH<4500 |
0.60>EUCIW>0.50 |
Fujian |
2503.3 |
2109.0 |
TAIWC<100 |
7500<IWCPH<12000 |
0.60>EUCIW>0.50 |
Jiangxi |
1996.7 |
2221.1 |
100<TAIWC<200 |
7500<IWCPH<12000 |
0.50>EUCIW>0.40 |
Shandong |
658.3 |
220.3 |
100<TAIWC<200 |
IWCPH<4500 |
EUCIW>0.60 |
Henan |
787.1 |
337.3 |
100<TAIWC<200 |
IWCPH<4500 |
EUCIW>0.60 |
Hubei |
1423.4 |
1498.0 |
100<TAIWC<200 |
4500<IWCPH<7500 |
0.60>EUCIW>0.50 |
Hunan |
1668.9 |
2196.6 |
100<TAIWC<200 |
7500<IWCPH<12000 |
0.60>EUCIW>0.50 |
Guangdong |
2357.6 |
2458.6 |
100<TAIWC<200 |
7500<IWCPH<12000 |
0.50>EUCIW>0.40 |
Guangxi |
1631.6 |
2178.6 |
100<TAIWC<200 |
IWCPH>12000 |
0.50>EUCIW>0.40 |
Hainan |
2341.5 |
489.9 |
TAIWC<100 |
IWCPH>12000 |
0.60>EUCIW>0.50 |
Chongqing |
1236.8 |
604.9 |
TAIWC<100 |
4500<IWCPH<7500 |
0.50>EUCIW>0.40 |
Sichuan |
921.3 |
2340.9 |
100<TAIWC<200 |
4500<IWCPH<7500 |
0.50>EUCIW>0.40 |
Guizhou |
1213.7 |
1066.1 |
TAIWC<100 |
4500<IWCPH<7500 |
0.50>EUCIW>0.40 |
Yunnan |
1295.9 |
2088.9 |
TAIWC<100 |
4500<IWCPH<7500 |
0.50>EUCIW>0.40 |
Tibet |
611.6 |
4642.2 |
TAIWC<100 |
7500<IWCPH<12000 |
0.50>EUCIW>0.40 |
Shaanxi |
626.2 |
271.5 |
TAIWC<100 |
IWCPH<4500 |
0.60>EUCIW>0.50 |
Gansu |
290.9 |
168.4 |
TAIWC<100 |
4500<IWCPH<7500 |
0.60>EUCIW>0.50 |
Qinghai |
304.7 |
612.7 |
TAIWC<100 |
7500<IWCPH<12000 |
0.50>EUCIW>0.40 |
Ningxia |
301.0 |
9.6 |
TAIWC<100 |
7500<IWCPH<12000 |
0.60>EUCIW>0.50 |
Notes: The administrative regions exclude Hong Kong, Macao, and Taiwan in this study.
References:
China Irrigation and Drainage Development Center 2016 China irrigation and drainage development report (in Chinese); China Irrigation and Drainage Development Center: Beijing, China, 2018.387-394.
Ministry of Water Resources of the People's Republic of China 2016 China water resources bulletin (in Chinese); Ministry of Water Resources of the People's Republic of China: Beijing, China, 2017.
Point 4: Finally, it seems to me that the conclusions are too general for such a detailed study at the regional level. Would it be possible to include more concrete measures?
Response 4: Thanks a lot for the comments. We have improved suggestions to make it more specific and concrete. We have put forward specific opinions on provinces with different hydro-economic conditions. For example:
……It is always good to be prepared, even in a province in which pressure on water resources is low, by improving the management of irrigation and the application and popularization of related technologies. The low-pressure pipeline water delivery irrigation is likely to be the most suitable technology for HE-1 provinces (low economic capacity, low hydrological challenges) because of its cost-effectiveness. For a province in which economic capacity is strong, more advanced technologies can be considered, such as the sprinkler irrigation and the micro irrigation. Considering that substantial regional hydrological challenges are the most important incentive or internal driving force to improve water use efficiency and water resources is so valuable in arid and semi-arid regions in northern China, efforts should be made continuously to develop highly efficient water-saving irrigation, such as the micro irrigation, despite the fact that some provinces have a weak economy. In addition to increasing capital investment in water-saving technology, it is very important to strengthen the belief in technological innovation and green development [75] and to build a policy and social environment that encourages technological innovation and water and energy conservation.
References:
Romer, P. M., Chapter 13 - Two strategies for economic development: Using ideas and producing ideas. In The Strategic Management of Intellectual Capital, Klein, D. A., Ed. Butterworth-Heinemann: Boston, 1998; pp 211-238.Reviewer 2 Report
The abstract must be improved. It should indicate the purpose of the study and the meaning of the abbreviations SSP1, SSP2, SSP3, SSP4, SSP5.
Line 159 - Table 1 - format header, so that the word Convergence stays on one line.
Review the units throughout the article. The units must be in the international system. For example: line 224; Table 2; Line 278, Table 5,….
In equation (3) the author must indicate the units of the variables.
Line 244 - t represents period, in line 245 t represents convergence time. The same acronym should not represent two things. Review the meaning of t.
Line 274 - Table 4 - Format Header. The same word must not be on two lines.
Figure 3 - In the caption correct the word corresponding. Improve font in subtitle text
Author Response
Response to Reviewer 2 Comments
Point 1: The abstract must be improved. It should indicate the purpose of the study and the meaning of the abbreviations SSP1, SSP2, SSP3, SSP4, SSP5.
Response 1: Thanks a lot for the comments. We have improved the abstract. The purpose of the study and the meaning of the abbreviations SSP1, SSP2, SSP3, SSP4, SSP5 have been added in the abstract. For example:
Abstract: Technological changes in water use efficiency directly influence regional sustainable development. However, few studies have attempted to predict changes in water use efficiency because of the complex influencing factors and regional diversity. The Chinese Government has established a target of 0.6 for the effective utilization coefficient of irrigation water but it is not clear how the coefficient will change in different provinces in future. The purpose of this study is to predict irrigation water use efficiency changes using a conditional convergence model and combined with the SSPs scenario settings and HE classification to group 31 Chinese provinces by their different economic and water resources conditions. The results show that the coefficient exponentially converges to 0.6 in half the provinces under SSP1 (sustainability), SSP2 (middle of the road) and SSP5 (conventional development) by 2030, whereas SSP3 (fragmentation) and SSP4 (inequality) are generally inefficient development pathways. HE-3 provinces (strong economic capacity, substantial hydrological challenges) achieve the greatest efficiency improvements (with all coefficients above 0.6), and SSP1 is a suitable pathway for these provinces. HE-2 provinces (strong economic capacities, low hydrological challenges) have relatively low efficiency because they lack incentives to save water, and SSP1 is also suitable for these provinces. For most HE-1 provinces (low economic capacity, low hydrological challenges), the coefficients are less than 0.6, and efforts are required to enhance their economic capacity under SSP1 or SSP5. HE-4 provinces (low economic capacity, substantial hydrological challenges) would improve efficiency in a cost-efficient manner under SSP2.
Point 2: Line 159 - Table 1 - format header, so that the word Convergence stays on one line.
Response 2: Thanks a lot for the comments. The format of table 1 has been revised. The word “convergence” stays on one line now.
Point 3: Review the units throughout the article. The units must be in the international system. For example: line 224; Table 2; Line 278, Table 5,….
Response 3: Thanks a lot for the comments. We have double checked and revised the data units in the manuscript. The monetary units have been converted into US dollars according to the 2016 average exchange rate in the 2016 statistical bulletin on national economic and social development released by the national bureau of statistics.
Point 4: In equation (3) the author must indicate the units of the variables. Line 244 - t represents period, in line 245 t represents convergence time. The same acronym should not represent two things. Review the meaning of t.
Response 4: Thanks a lot for the comments. We have added units of the variables; we have double checked and revised the meaning of t, which represents convergence time. For example:
…where (scalar) represents water use efficiency in convergence time (years), (scalar) represents water use efficiency for medium- to long-term targets, (scalar) represents initial water use efficiency in a region, and (scalar) represents the convergence control parameters in a specific region.
Point 5: Line 274 - Table 4 - Format Header. The same word must not be on two lines.
Response 5: Thanks a lot for the comments. The format of table 4 has been revised. The same word stays on one line now.
Point 6: Figure 3 - In the caption correct the word corresponding. Improve font in subtitle text
Response 6: Thanks a lot for the comments. The word “corresponding” in the caption has been corrected. We have improved font in subtitle text by using “Palatino Linotype” font. Also, we have improved the quality of figure 3. Figure 3 is at a resolution of 600 dpi now.
Reviewer 3 Report
The article is very well structured and approaches actual issues on water use efficiency and sustainable development. The introduction is sufficiently well developed and includes relevant information to the other chapters of the article. This article is based on a methodology which leads to very interesting results. However, there are some shortcomings which must be approached by the authors.
1. Tables 3 and 4 should be better explained considering that they should provide clear answers to one of the article objectives
2. The figures which support these results could be improved. I must remark the poor quality of figure 3.
3. The suggestions list proposed for improving the irrigation water use efficiency of each province should be improved/ extended from a more practical view giving the importance of the approached subject.
Author Response
Response to Reviewer 3 Comments
Point 1: Tables 3 and 4 should be better explained considering that they should provide clear answers to one of the article objectives
Response 1: Thanks a lot for the comments. Tables 3 and 4 have been better explained. For example:
…SSP2 is a medium scenario in every HE classification; SSP5 presents high efficiency in every HE classification. SSP2 is also more pessimistic about the target of convergence than SSP5 and has shorter convergence time than SSP5…
Point 2: The figures which support these results could be improved. I must remark the poor quality of figure 3.
Response 2: Thanks a lot for the comments. We have improved the quality of all figures. All figures are at a resolution of 600 dpi now. The format of figure 3 has been revised. The word “corresponding” in figure 3 has been corrected. We have improved font in text in figure 3 by using “Palatino Linotype” font.
Point 3: The suggestions list proposed for improving the irrigation water use efficiency of each province should be improved/ extended from a more practical view giving the importance of the approached subject.
Response 3: Thanks a lot for the comments. We have improved suggestions to make it more specific and concrete. We have put forward specific opinions on provinces with different hydro-economic conditions. For example:
……It is always good to be prepared, even in a province in which pressure on water resources is low, by improving the management of irrigation and the application and popularization of related technologies. The low-pressure pipeline water delivery irrigation is likely to be the most suitable technology for HE-1 provinces (low economic capacity, low hydrological challenges) because of its cost-effectiveness. For a province in which economic capacity is strong, more advanced technologies can be considered, such as the sprinkler irrigation and the micro irrigation. Considering that substantial regional hydrological challenges are the most important incentive or internal driving force to improve water use efficiency and water resources is so valuable in arid and semi-arid regions in northern China, efforts should be made continuously to develop highly efficient water-saving irrigation, such as the micro irrigation, despite the fact that some provinces have a weak economy. In addition to increasing capital investment in water-saving technology, it is very important to strengthen the belief in technological innovation and green development [75] and to build a policy and social environment that encourages technological innovation and water and energy conservation.
References:
Romer, P. M., Chapter 13 - Two strategies for economic development: Using ideas and producing ideas. In The Strategic Management of Intellectual Capital, Klein, D. A., Ed. Butterworth-Heinemann: Boston, 1998; pp 211-238.